# Comparing modelling approaches for the estimation of government intervention effects in COVID-19: Impact of voluntary behavior changes

Lun Liu[1,2], Zhu Zhang[1], Hui Wang[3]*, Shenhao Wang[4,5,6], Shengsheng Zhuang[7], Jishan Duan[8]

1 School of Government, Peking University, Beijing, China, 2 Institute of Public Governance, Peking University, Beijing, China, 3 School of Architecture, Tsinghua University, Beijing, China, 4 Department of Urban and Regional Planning, University of Florida, Gainesville, Florida, United States of America, 5 Department of Urban Studies and Planning, Massachusetts Institute of Technology, Cambridge, Massachusetts, United States of America, 6 Media Lab, Massachusetts Institute of Technology, Cambridge, Massachusetts, United States of America, 7 Peking Union Medical College, Beijing, China, 8 Graduate School of Architecture, Planning and Preservation, Columbia University, New York, New York, United States of America

* wh-sa@mail.tsinghua.edu.cn

**Data Availability Statement:** All codes for analysis and visualization presented in this manuscript is available at https://github.com/lunliu454/covid19_

## Abstract

The efficacy of government interventions in epidemic has become a hot subject since the onset of COVID-19. There is however much variation in the results quantifying the effects of interventions, which is partly related to the varying modelling approaches employed by existing studies. Among the many factors affecting the modelling results, people's voluntary behavior change is less examined yet likely to be widespread. This paper therefore aims to analyze how the choice of modelling approach, in particular how voluntary behavior change is accounted for, would affect the intervention effect estimation. We conduct the analysis by experimenting different modelling methods on a same data set composed of the 500 most infected U.S. counties. We compare the most frequently used methods from the two classes of modelling approaches, which are Bayesian hierarchical model from the class of computational approach and difference-in-difference from the class of natural experimental approach. We find that computational methods that do not account for voluntary behavior changes are likely to produce larger estimates of intervention effects as assumed. In contrast, natural experimental methods are more likely to extract the true effect of interventions by ruling out simultaneous behavior change. Among different difference-in-difference estimators, the two-way fixed effect estimator seems to be an efficient one. Our work can inform the methodological choice of future research on this topic, as well as more robust re-interpretation of existing works, to facilitate both future epidemic response plans and the science of public health.

intervention_model_compare. This work is licensed under a Creative Commons Attribution 4.0 International (CC BY4.0) license, which permits unrestricted use, distribution, and reproduction in any medium, provided the original work is properly cited. To view a copy of this license, visit https://creativecommons.org/licenses/by/4.0/.

**Funding:** This work is supported by the Beijing Social Science Foundation (20GLA003, L.L., http://www.bjsk.org.cn), the Tsinghua University Spring Breeze Fund (2021Z99CFY038, H.W., https://www.tsinghua.edu.cn), the National Natural Science Foundation of China (52008005, L.L., https://www.nsfc.gov.cn), the National Social Science Fund of China (19FGLB069, L.L., http://www.nopss.gov.cn) and the Institute of Public Governance, Peking University (YQZX202005, TDXM202104, L.L., http://www.ggzl.pku.edu.cn). The funders had no role in study design, data collection and analysis, decision to publish, or preparation of the manuscript.

**Competing interests:** The authors have declared that no competing interests exist.

## Introduction

In fighting COVID-19, non-pharmaceutical interventions aiming at reducing mobility and contact have been implemented by governments around the world repetitively. Given the strong (and mostly negative) impact of these interventions on economy and social life, the efficacy of interventions has become a hot subject of study [1–5], which is informative for both future epidemic response plans and the science of public health. However, there is much variation in the analysis results quantifying the effects of the interventions, which prevents researchers and policy makers from drawing clear and reliable conclusions (estimates with the reproduction number as the outcome of interest are summarized in S1 Table). There are two major sources of such variation: the first is the sample data (countries, regions) used in the analyses, as the effects of interventions might be naturally different in different local contexts influenced by the intensity of enforcement, lifestyle, culture and so on; the second is the modelling approach, which might capture different structures in the data thus needs careful evaluation. This paper therefore aims to analyze how the choice of modelling approach would affect the intervention effect estimation.

While the impact of model features such as epidemiological assumptions have been extensively analyzed [6], the impact of voluntary behavior change is only mentioned yet not examined. Voluntary behavior changes refer to people's spontaneous self-protection such as wearing masks and avoiding contacts in face of the epidemic [7, 8], whether or not there are government interventions. Such behavior is likely to increase when infections are high, which is also the case for government interventions, resulting in potential confusion in intervention effect identification. For instance, Brauner et al. (2021) mention that reproduction numbers may be reduced by voluntary behavior changes, which is a topic to explore further.

While mathematical models (such as SIR and SEIR models) are classical methods in modelling epidemics [9, 10], statistical models and their combinations are widely applied in estimating the effects of interventions. Relevant statistical models employed by existing analyses can be grouped into two classes: those that incorporate a natural experimental design and those that do not (review of the methods used in 23 highly relevant studies in S1 Table). The former usually employ tools from the field of econometrics or biostatistics which have been developed for identifying causal influences in public policy and medical issues. The natural experimental design is featured by comparing regions that implemented or relaxed an intervention in a period (treatment group) to regions that did not (control group), given that the latter group is similar enough to the former group to be considered as the counterfactual. The dominant method used has been difference-in-difference [4, 11–16], a method that identifies causal treatment effect through comparing pre-and post-treatment performance of control and treatment groups [17]; and occasionally synthetic control [18] and interrupted time series analysis [19]. The second class of analyses do not incorporate a treatment-control comparison but focus on maximizing model fit to observed data based on assumed model structures. Specific methods that have been used include Bayesian hierarchical model [2, 3, 20–24], generalized linear regression [5, 25–30] and certain machine learning models [5].

Both groups of modelling methods are with merits. The assumed advantage of natural experimental design lies in the ability to rule out common temporal changes in the treatment and control groups, which might otherwise be mistaken as the treatment effect, thus identifying the true effect of an intervention. To be more specific, if there is a natural trend in the epidemic such as voluntary behavior changes happening together with an intervention effect, researchers might be able to rule it out by subtracting the epidemic course of the treatment group with that of the control group [14]. On the other hand, the more computational methods are better at modelling nonlinear and inconstant relationships between interventions, epidemiological parameters and infection outcomes [6].

This paper therefore aims to empirically examine how the choice of modelling approach would affect the estimation results of intervention effects, by experimenting with different modelling approaches on a same data set. We focus on the most frequently used methods from the two classes, which are difference-in-difference from the class of natural experimental methods and Bayesian hierarchical model from the class of more computational methods (can be checked in S1 Table). It should be noted that among the non-natural experimental methods, generalized linear regression is also frequently used, however, most of the model settings are not able to capture the causal relationships and only association can be claimed from the analysis results [25, 26, 29, 31]. More specifically, in terms of Bayesian hierarchical model, we use the model developed by Brauner et al. (2021), which is a later modification in a group of similar models [3, 24]. This model uses infection case and death data in each region to backward infer daily reproduction numbers and then the impacts of interventions on the reproduction number. In terms of the difference-in-difference method, we test two estimators. The first is the two-way fixed effect estimator, which is a widely used difference-in-difference estimator in policy analysis and the most used one in relevant studies [17]. The estimation is implemented through a linear model with the outcome of interest as dependent variable and intervention status per region per day as well as region and day fixed effects as independent variables. Estimating the linear model is equivalent to comparing the changes in treatment and control groups before and after intervention when there are two groups and two time periods. However, this estimator could be biased if there are multiple time periods and the treatment effect changes over time, which is possible in the case of COVID-19 as the compliance to interventions may change (e.g. it may take some time for compliance to increase, and compliance may also decrease as time goes) [24, 32]. To accommodate the heterogeneous effect, we further experiment with a new difference-in-difference estimator that is robust to temporally heterogeneous treatment effect [33, 34]. The robust estimator directly compares the epidemic course after $n$ days of intervention status change in all possible pairs of treatment and control groups across the entire study period and computes the average treatment effect ($n \geq 0$).

We take the United States as the case and analyze the effect of government interventions using county-level data of 500 counties with the largest number of infections. We use data from the first pandemic wave, that is, from March to August 2020, since intervention effect estimates in later periods could be confounded by more factors including lockdown fatigue, virus variants and vaccination. Six widely applied government interventions are evaluated, which are stay-at-home order, school closure, childcare closure, non-essential retail closure, banning small-size gatherings (below 10 people), and banning large-size gatherings (above 10 people) (Fig 1).

Accurate knowledge of the effectiveness of government interventions is key to cost-effective policy making not only for controlling the on-going COVID-19 but also for future public health crisis. By directly experimenting with three main stream estimation methods, this work aims to facilitate a better understanding on the strength and weakness of relevant modelling approaches and the potential bias of reported results. While we cannot fully encompass all possible methodology, our analysis is nonetheless informative on refining the analysis on this issue and grounding decision making on sound scientific evidences.

## Method

### Data

The analysis involves daily infection case and death data, county-level mobility data, and government intervention data on 500 counties with the most infection cases in the United States between March 13 and August 15 2020. The infection case and death data is downloaded from

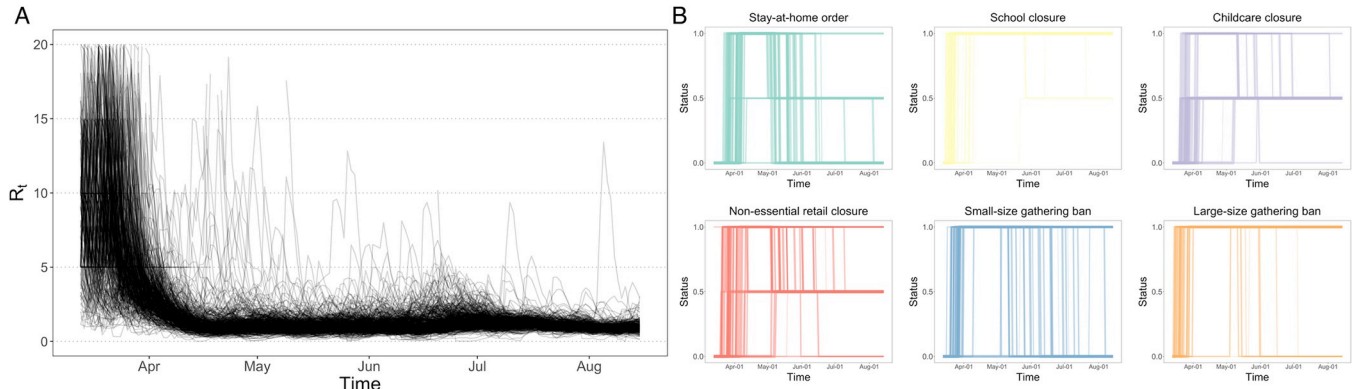

**Fig 1. Epidemic course and interventions in the sample counties in the United States. (A)** Dynamics of daily instantaneous reproduction numbers ($R_t$) in the sample counties. The reproduction numbers were high at the beginning of the pandemic, decreased in April and slightly upsurged in July. Nonetheless, there was much variation in the reproduction numbers across counties. **(B)** Dynamics of the six interventions. The interventions switched between none (0), partial restriction (0.5) and full restriction (1) during the study period. The widths of lines in the graphs are proportional to the number of counties in the corresponding status on the corresponding day. Since interventions were decided by individual states, the timing differed a lot across the sample counties, based on which the intervention effects could be estimated.

the COVID-19 Data Repository by the Center for Systems Science and Engineering (CSSE) at Johns Hopkins University [35]. The information on people's mobility is from Google's COVID-19 Community Mobility Reports [36]. The time of implementing and relaxing interventions is collected manually from the official websites of state governments, which is the major level of government in charge of intervention policy making in COVID-19 in the United States. We do not account for the few cases in which county or local governments take alternative actions than the orders of state governments, which are only occasional [12]. The interventions are coded 0, 0.5 or 1 based on the intervention status in each region on a given day: 0 if there was no relevant restriction, 1 if a full restriction was implemented, and 0.5 if there was a partial restriction (e.g. recommending instead of enforcing a restriction, shortening opening hours or limiting occupancy instead of a complete closure).

## Testing common behavioral trends

We examine whether there is a common behavioral trend across all sample counties using Google's COVID-19 Community Mobility Reports. This dataset provides the daily percentage change in the visits to different categories of places (or the time spent at places of residence) comparing with baselines (different baselines for each day of the week, using the median value from the 5-week period Jan 3 to Feb 6, 2020). We focus on the 'residence' indicator, referring to the percentage change in the time spent at home, since it can reflect the overall change of mobility. Next, we test whether there is a 'common component' in the time people staying at places of residence each day across counties after controlling for interventions, by regressing the Google mobility indicator against the daily status of interventions in each county and a series of binary variable indicating each day of the study period (i.e. day-fixed effect). If there is a day effect, then the coefficient for the day variable should be significantly different from zero. The regression model is expressed as follows

$$m_{c,t} = \sum_{i \in I} \beta_i x_{i,c,t} + \gamma_t + e_{c,t} \qquad (1)$$

where $m_{c,t}$ denotes the Google mobility indicator in county $c$ on day $t$; $x_{i,c,t}$ denotes the status of intervention $i$ in county $c$ on day $t$ and $\beta_i$ denotes coefficients; $I$ is the set of interventions

being studies; $\gamma_t$ denotes day $t$, representing the common part of mobility change across counties; $e_{c,t}$ is the error term. $\beta_i$ and $\gamma_t$ are estimated using ordinary least square method.

## Experiment 1. Bayesian hierarchical model

We use the model developed by Brauner et al. (2021), which is an extension of the original model proposed by Flaxman et al. (2020). The semi-mechanistic model is composed of a series of equations that link intervention status to instantaneous reproduction numbers, and then to the total number of infections and the number of observed cases and deaths. The key equations are as follows.

$$R_{t,c} = R_{0,c} \prod_{i=1}^{I} exp(-\alpha_{i,c} x_{i,t,c}) \tag{2}$$

$$R_{t,c} = \frac{1}{M(-\log(1 + g_{t,c}))} \tag{3}$$

$$N_{t,c}^{(C)} = N_{o,c}^{(C)} \prod_{\tau=1}^{t} [(1 + g_{\tau,c}) \cdot exp(\varepsilon_{\tau,c}^{(C)})] \tag{4}$$

$$N_{t,c}^{(D)} = N_{o,c}^{(D)} \prod_{\tau=1}^{t} [(1 + g_{\tau,c}) \cdot exp(\varepsilon_{\tau,c}^{(D)})] \tag{5}$$

$$\bar{y}_{t,c}^{(C)} = \sum_{\tau=0}^{31} N_{t-\tau,C}^{(C)} P_C(delay = \tau) \tag{6}$$

$$\bar{y}_{t,c}^{(D)} = \sum_{\tau=0}^{47} N_{t-\tau,C}^{(D)} P_D(delay = \tau) \tag{7}$$

where $R_{o,c}$ and $R_{t,c}$ denote the basic reproduction number in county $c$ and the instantaneous reproduction number on day $t$ in county $c$; $x_{i,t,c}$ denotes the status of intervention $i$ in county $c$ on day $t$ and $I$ is the number of interventions; $\alpha_{i,c}$ denotes the effect of intervention $i$ in county $c$, which is assumed to be county-specific and independent multiplicative on $R_{t,c}$. $\alpha_{i,c}$ is assumed to follow a normal distribution $\alpha_{i,c} \sim$ Normal($\alpha_i$, $\sigma^2_i$) and the prior distributions for $\alpha_i$ and $\sigma_i$ are $\alpha_i \sim$ Asymmetric Laplace ($m = 0$, $\kappa = 0.5$, $\lambda = 10$) and $\sigma_i \sim$ Half Student-T ($\nu = 3$, $\sigma = 0.04$). In Eq 3, $g_{t,c}$ denotes the daily growth rate of infections and M($\cdot$) is a moment-generating function; the equation can be solved with information on the generation interval, which is the time between successive infections in a transmission chain. In Eqs 4 to 7, $N^{(C)}_{t,c}$ and $N^{(D)}_{t,c}$ denote the number of infections that subsequently become confirmed cases or reported deaths, respectively; $g_{\tau,c}$ is the daily growth rate; $\varepsilon^{(.)}_{\tau,c}$ is a noise term; $P_C(delay)$ and $P_D(delay)$ are the distribution of the delay from infection to confirmation and from infection to death, respectively. $\varepsilon^{(.)}_{\tau,c}$ is assumed to follow a normal distribution $\varepsilon^{(.)}_{\tau,c} \sim$ Normal(0, $\sigma_N$) and a prior distribution is placed on $\sigma_N \sim$ Half Student-T ($\nu = 3$, $\sigma = 0.15$). $N^{(C)}_{0,c}$ and $N^{(D)}_{0,c}$ are placed with uninformative priors. The model estimates the posterior distributions of the parameters, including the effects of interventions ($\alpha_{i,c}$), using a Markov Chain Monte Carlo sampling algorithm. More details about the model can be found in Brauner et al. (2021).

## Experiment 2. Difference-in-difference: Two-way fixed effect estimator

Two-way fixed effect model is a widely used modelling method to implement difference-in-difference analysis [17]. The model regresses the outcome of interest on the treatment variable and at the same time, controls for unit fixed effects and time fixed effects, the latter of which helps rule out the changes happening simultaneously in intervention groups and control groups at each time point. The model is expressed as follows.

$$\log(R_{c,t}) = \sum_{i \in I} \beta_i x_{i,c,t} + \alpha_c + \tau_t + \varepsilon_{c,t} \tag{8}$$

where $log(R_{c,t})$ denotes the log-transformed instantaneous reproduction number in county $c$ on day $t$; $x_{i,c,t}$ denotes the status of intervention $i$ in county $c$ on day $t$ and $\beta_i$ denotes their coefficients; $\alpha_c$ and $\tau_t$ denote the unit (county) and day fixed effects, respectively; and $\varepsilon_{c,t}$ is the error term. The parameters are estimated with ordinary least squares. We estimate robust standard errors in a way that allows $\varepsilon_{c,t}$ to correlate at the county level [37]. More details of the method can be found in Angrist and Pischke (2008) [17].

We use $log(R_{c,t})$ as the dependent variable to match with the Bayesian hierarchical model, whose parameter estimation is interpreted as the proportional change of $R_t$ by interventions. The coefficients of intervention variables in this model can be interpreted in the same way by taking $exp(\beta_i)$. Daily $R_t$ in each county is estimated with the method proposed by Cori et al. [38], with a 7-day sliding time window to smooth infection numbers and parameters of the serial interval of infections based on previous epidemiological investigation of COVID-19 (mean = 7.5 days, standard deviation = 3.4 days) [39]. $R_t$ estimates with coefficient of variation greater than 0.3 are excluded from further analysis (mainly due to small case numbers in the corresponding time window).

## Experiment 3. Difference-in-difference: Robust estimator

The heterogeneity robust difference-in-difference estimator is proposed by de Chaisemartin and D'Haultfoeuille [33]. This estimator estimates the average treatment effect of implementing an intervention for $k$ days ($k \geq 0$) across all the regions where an intervention switched from level $d$ ($d \in \{0, 0.5, 1\}$) on $t\text{-}1$ to $d^{'}$ ($d^{'} \in \{0, 0.5, 1\}$, $d^{'} \neq d$) on $t$ and remained $d^{'}$ at least till day $t+k$ (switching regions), using the regions where the intervention remains $d$ between $t\text{-}1$ and $t+k$ as controls (stable regions). Since the treatment effect is estimated through explicit matching between switching and stable regions and analyzing their difference after a given number of days of treatment, temporal heterogeneity will not create bias. The estimator is computed as follows

$$DID_{i,k} = \sum_{(t,d,d'):t \geq 2, t \leq T-k, (d,d') \in D^2, d \neq d'} \frac{N_{i,d,d',t,k} DID_{i,d,d',t,k}}{N_s(d' - d)} \tag{9}$$

$$DID_{i,d,d',t,k} = \sum_{c:D_{i,c,t+k}=\ldots=D_{i,c,t}=d', D_{i,c,t-1}=d, t \geq 2} \frac{e_{i,c,t,k}}{N_{i,d,d',t,k}} - \sum_{c:D_{i,c,t+k}=\ldots=D_{i,c,t}=D_{i,c,t-1}=d, t \geq 2} \frac{e_{i,c,t,k}}{N_{i,d,d,t,k}} \tag{10}$$

$$e_{i,c,t,k} = \log(R_{c,t+k}) - \log(R_{c,t-1}) - (X'_{i,c,t+k} - X'_{i,c,t-1})\gamma - b \tag{11}$$

where $DID_{i,k}$ denotes the estimated effect of intervention $i$ after $k$ days of implementation; $t \in \{1,\ldots,T\}$ denotes the days after implementation; $N_s$ denotes the total number of counties where intervention status switched; $N_{i,d,d',t,k}$ and $N_{i,d,d,t,k}$ denote the number of switching counties and stable counties between day $t$ and $t+k$. If either $N_{i,d,d',t,k}$ or $N_{i,d,d,t,k}$ is 0, then $DID_{i,d,d',t,k}$

is defined as 0. $D_{i,c,t}$ denotes the status of intervention $i$ in county $c$ on day $t$; $e_{i,c,t,k}$ denotes the change in the logarithm of instantaneous reproduction number in county $c$ between day $t$ and $t+k$ after controlling for the impact of other interventions $X'_{i,c,t}$; $\gamma$ is a vector of coefficients and $b$ is the constant. $\gamma$ and $b$ are estimated through ordinary least squares using only stable observations. The standard error of $DID_{i,k}$ is estimated by bootstrapping for 100 times. Further, since there could be correlation of errors at the day level [4, 40], we utilized block bootstrap with days as blocks. More details of the method can be found in de Chaisemartin and D'Haultfoeuille (2020) [33].

To compare with the estimates obtained with the other two methods, we further compute the average intervention effect across $k$ days and the corresponding standard error. The average effect is taken as the mean of $DID_{i,0}$ to $DID_{i,k}$ and the standard error is computed as follows.

$$se_{i,0-k} = \sqrt{\frac{\sum_{a=0}^{k} se_{i,a}^2 + \sum_{b=0,b\neq a}^{k} \sum_{a=0}^{k} cov_{i,a,b}}{(k+1)^2}} \tag{12}$$

where $se_{i,0-k}$ denotes the standard error of the average effect after $k$ days of intervention $i$, $se_{i,a}$ denotes the standard error of $DID_{i,a}$ ($a$ takes values between 0 and $k$); and $cov_{i,a,b}$ denotes the covariance between $DID_{i,a}$ and $DID_{i,b}$. We estimate $DID_{i,k}$ for three weeks after an intervention, which is a common length of interventions in practice.

## Result

### Existence of common behavioral trends

We start by testing whether common behavioral trends exist in the country and whether they correlate with the implementation or relaxation of interventions. Among different self-protection behaviors, here we focus on residents' mobility reduction, which is easier to monitor through personal location data than other behaviors such as wearing masks. On this aspect, Google's COVID-19 Community Mobility Reports provide regional-level indicators on residents' daily mobility (at the county level for the United States) [36].

By regressing the county-specific daily mobility indicators on day variables and intervention status, we find a statistically significant day effect on most days in our study period, indicating a non-zero common change in the amount of travel conducted by residents in different counties on each day (Fig 2). There was first an upward common trend in the time staying at home from March to mid-April, followed by a gradual decrease afterwards, yet by the end of the period, the time staying at home was still higher than before the pandemic. Actually, the day effects account for a larger proportion of the total variance in mobility than government interventions, as the adjusted $R^2$ of the model is 0.22 with only intervention status as explanatory variables and increases to 0.66 when day effects are added (full results in S2 Table). Correlation tests show that this common behavioral trend is statistically significantly correlated with the status of interventions (Pearson's $r = 0.02\sim0.39$, all statistically significant at 0.001 level). The positive correlation suggests that the estimation of intervention effects could be upwardly biased by simultaneous voluntary behavior changes without proper controlling measures, which confirms the concern behind natural experimental methods.

### Comparison of model results

The intervention effect estimates produced by the three methods are shown in Fig 3, which are different from each other given the diverse modelling strategies. Generally speaking, the semi-

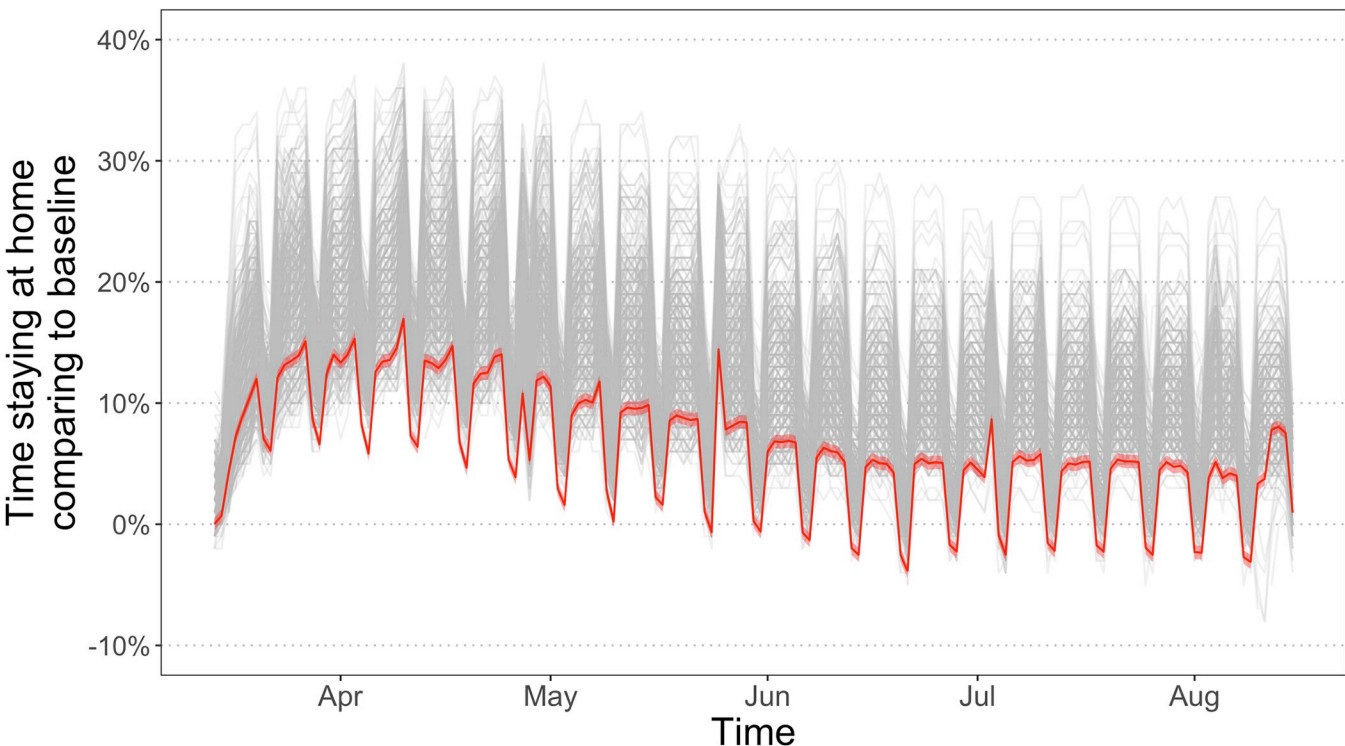

**Fig 2. Trend of the common change in residents' mobility.** Grey lines are the daily values of the Google mobility indicator in the sample counties and the red line is the common daily mobility change shared by the counties with 95% confidence interval. The red line demonstrates that there is a non-zero common trend of mobility across counties. Please note that the baseline in the Google mobility indicator is the median value on each day of the week from the 5-week period Jan 3 to Feb 6, 2020, in other words, different baselines are used for different days of the week.

mechanistic model produces the largest estimates on interventions' effects in reducing $R_t$, which is consistent with our reasoning that non-natural experimental methods are likely to produce larger estimates. Three interventions are estimated to have a statistically significant impact on reducing $R_t$, which are school closure (reducing $R_t$ by -67.4%, 95% confidence interval: -69.7 to -64.8%), childcare closure (reducing $R_t$ by -17.6%, -24.5 to -9.7%), and non-essential retail closure (reducing $R_t$ by -6.9%, -14.9 to -1.2%). The particularly large effect of school closure might be related to the interaction between the sequence of interventions and structure of the model [41].

However, none of these three interventions are found to have a statistically significant impact according to the two difference-in-difference estimators, which could be a consequence of excluding voluntary behavior changes by the natural experiment design. Generally, estimation results acquired from the two difference-in-difference estimators are closer to each other than to the results of the semi-mechanistic model (differences are not statistically significant). Stay-at-home order is found to have the most prominent effect in reducing $R_t$ by both of the two difference-in-difference estimators: -6.6% (-10.6 to -2.5%) as estimated by the two-way fixed effect estimator and -17.3% (-31.7 to 0%) as estimated by the robust estimator, which nonetheless are not statistically different (z-statistics = 1.27). The wider confidence interval of the latter could be because the estimation is on a day-by-day basis so that a smaller sample is used in estimating the impact of implementing or relaxing an intervention for $n$ days ($n \in \{0 \sim 21\}$), resulting in more uncertainties. Besides, banning small-size gatherings is also found to statistically significantly reduce $R_t$ by the two-way fixed effect estimator (-5.1%, -9 to -1.1%), while its estimate is associated with a wide confidence interval when using the robust

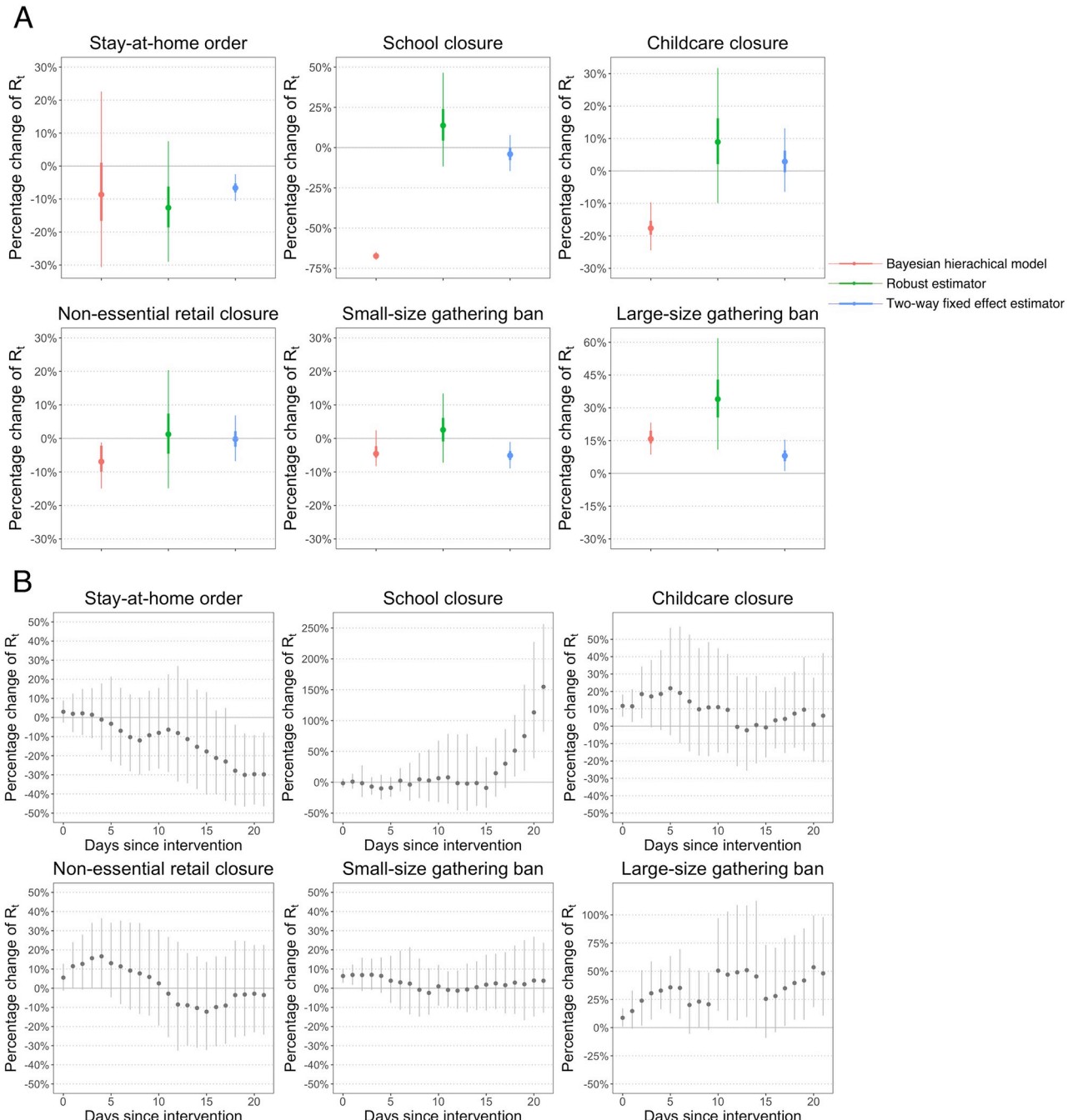

**Fig 3. Intervention effect estimates from different modelling approaches. (A)** Comparison of the results from the three approaches, with 50% and 95% confidence intervals. The Bayesian hierarchical model tends to produce the larger effect estimates, since the effects of simultaneous voluntary behavioral changes could be taken by the model. The results produced by the two difference-in-difference estimators are closer to each other, yet the robust estimator has wider confidence intervals. **(B)** Time-varying intervention effects estimated by the robust difference-in-difference estimator with 95% confidence intervals. It shows that stay-at-home order has an increasing effect over time.

estimator. The other interventions are not found to have a statistically significant impact by both estimators. All the interventions satisfy the parallel pre-trend assumption, which means that the epidemic course in the control group is similar to that of the treatment group before

intervention status changes, so that the former can be proper counterfactuals for the latter (details on the methodology and results of parallel pre-trend test in Supplementary Method and S3 & S4 Tables).

Since the difference between the results from the two difference-in-difference estimators should stem from the temporally heterogeneous intervention effects, we then examine the day-by-day estimates of intervention effects produced by the robust difference-in-difference estimator (Fig 3B). The results suggest that the intervention effects do change with the length of implementation. Among the two interventions that are found to have significant impacts on reducing $R_t$ in the two-way fixed effect model, stay-at-home order shows an increasing effect over time, which might be caused by the gradual change of behavior and enforcement measures; and small-size gathering ban demonstrates a V-shaped trend, suggesting a loss of effect after initial effectiveness.

## Discussion

The wide use of non-pharmaceutical interventions in COVID-19 provides large-scale natural experiments on the impacts of various public health measures, which are valuable information for fostering the knowledge in relevant fields such as public health, urban resilience, public policy and informing future policy making. Diverse methodologies have been developed to evaluate the efficacy of interventions in COVID-19. By testing three major modelling methods on the epidemic data from the United States, we demonstrate that quite different estimates of intervention effects could be acquired with different methods. Particularly, methods without explicit comparison of treatment and control groups tend to over-estimate intervention effects, as we find there is non-negligible voluntary behavior changes happening together with the interventions. This is further testified by comparing our estimates to previous analyses—our estimates from difference-in-difference methods are generally smaller than estimates from non-natural experimental methods [2, 22, 25, 28], though this comparison is not absolutely valid due to varying geographical extents of the studies. The results suggest that estimates produced by methods with a natural experiment design could be more reliable, despite of differences caused by variance in details of modelling approaches. Our work can inform the methodological choice of future research on this topic as data are still accumulating, and also facilitate more robust re-interpretation of existing works.

Regarding to the two estimators with natural experimental design, the one robust to temporal heterogeneity in intervention effects is methodologically more superior. However, since much less observations are available for each $n$-day effect estimate, there could be less certainty about the estimate thus wider confidence intervals are produced. Considering that the results from the two difference-in-difference estimators are not statistically significantly different, the two-way fixed effect estimator is a preferable choice, which also has the advantage of fast computing. On our data, it takes 42 CPU hours (2.6 GHz) to finish bootstrapping 100 times for the estimation of one intervention. Nonetheless, if longer period data is collected and used, the robust estimator might be able to draw narrower confidence intervals thus become a more preferable choice.

Nonetheless, when employing the two-way fixed effect estimator, one should be aware of the possible bias, which is jointly influenced by the intervention effect in each treatment-control pair and their distribution across time [33, 34]. Therefore, the difference between the results of the two difference-in-difference estimators in our analysis might not hold for other countries and time periods thus could not be taken as a reference. Neither might the temporal trends of intervention effects in this analysis (Fig 3B) hold in alternative contexts, which could be influenced by the dynamics of enforcement intensity and compliance.

A limitation to our methodological analysis could be the potential signaling or spillover effect of interventions. The former refers to the possibility that increased intervention in treatment regions makes people in the control group alerted and more cautious, or the reversed in the reopening stage [14]. The latter refers to the possibility that intervention in treatment regions also reduces (or increases in reopening) infections in the control group due to change in cross-region travel [12]. Both cases would deflate the intervention effect estimates in all the methods.

Although significant bias is found with non-natural experimental models in estimating intervention effects, this can be modified by including model elements that simulate voluntary behavior changes in response to the severity of epidemic. Our conclusions on modelling approach comparison may also apply to the evaluation of intervention effects in other areas which involve massive behavior changes such as disaster recovery and crime mitigation.

## Supporting information

**S1 File. Supplementary methods.**
(DOCX)

**S1 Table. Methodological summary of studies on COVID-19 intervention effect.** Sample and results are provided if the study takes Rt as the outcome of interest, which are comparable to our analysis.
(DOCX)

**S2 Table. Results of models on intervention status and residents' mobility.**
(DOCX)

**S3 Table. Pre-trend test for the two-way fixed effect estimator.**
(DOCX)

**S4 Table. Pre-trend test for the robust difference-in-difference estimator.**
(DOCX)

## Acknowledgments

We thank the High-performance Computing Platform of Peking University for providing the computation resource.

## Author Contributions

**Conceptualization:** Lun Liu, Hui Wang.

**Data curation:** Zhu Zhang.

**Formal analysis:** Lun Liu, Zhu Zhang, Shengsheng Zhuang, Jishan Duan.

**Funding acquisition:** Lun Liu, Hui Wang.

**Investigation:** Lun Liu, Zhu Zhang.

**Methodology:** Lun Liu.

**Writing – original draft:** Lun Liu, Hui Wang.

**Writing – review & editing:** Lun Liu, Hui Wang, Shenhao Wang.

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
