## [Decision Letter · Decision Letter 0]

23 Jun 2022

PONE-D-22-07518Comparing modelling approaches for the estimation of government intervention effects in COVID-19: Impact of voluntary behavior changesPLOS ONE

Dear Dr. Wang,

Thank you for submitting your manuscript to PLOS ONE. After careful consideration, we feel that it has merit but does not fully meet PLOS ONE’s publication criteria as it currently stands. Therefore, we invite you to submit a revised version of the manuscript that addresses the points raised during the review process.

We look forward to receiving your revised manuscript.

Kind regards,

Lei Shi

Academic Editor

PLOS ONE

Journal Requirements:

This work is supported by the Beijing Social Science Foundation (20GLA003, L.L.), the Tsinghua University Spring Breeze Fund (2021Z99CFY038, H.W.), the National Natural Science Foundation of China (52008005, L.L.) and the Institute of Public Governance, Peking University (YQZX202005, L.L.). We thank the High-performance Computing Platform of Peking University for providing the computation resource.

However, funding information should not appear in the Acknowledgments section or other areas of your manuscript. We will only publish funding information present in the Funding Statement section of the online submission form. 

This work is supported by the Beijing Social Science Foundation (20GLA003, L.L., http://www.bjsk.org.cn), the Tsinghua University Spring Breeze Fund(2021Z99CFY038, H.W., https://www.tsinghua.edu.cn), the National Natural Science Foundation of China (52008005, L.L., https://www.nsfc.gov.cn) and the Institute of Public Governance, Peking University (YQZX202005, L.L., http://www.ggzl.pku.edu.cn). The funders had no role in study design, data collection and analysis, decision to publish, or preparation of the manuscript.

Reviewers' comments:

Reviewer's Responses to Questions

**Comments to the Author**

1. Is the manuscript technically sound, and do the data support the conclusions?

Reviewer #1: Yes

Reviewer #2: Yes

2. Has the statistical analysis been performed appropriately and rigorously? 

Reviewer #1: Yes

Reviewer #2: Yes

3. Have the authors made all data underlying the findings in their manuscript fully available?

Reviewer #1: Yes

Reviewer #2: Yes

4. Is the manuscript presented in an intelligible fashion and written in standard English?

Reviewer #1: Yes

Reviewer #2: Yes

5. Review Comments to the Author

Reviewer #1: By testing three major modeling methods, they demonstrate that quite different estimates of intervention effects could be acquired with different methods. The idea of the model seems somehow convincing and worth reporting. However, the way the current version of the manuscript is presented cannot be accepted and needs a major revision. Here are a couple of points that I think should be considered.

Critical remarks:

(1) The model depiction is not clear to me. The language in explaining the model should be improved. In particular, I cannot find how to get indicators ‘shared change of mobility’ and ‘value of the mobility indicator in county c on day t.

(2) In Experiment 1, I suggest the authors give some details in the appendix. At least, the algorithm and the details should be given. In Experiment 2 and Experiment 3, the process of solving is not clear.

(3) The reason why the authors choose these methods is not clear. In reference to what you give, their topics are all related to COVID-19. Thus, the innovative point of this paper should be emphasized. What is the difference between your content and theirs that should be given?

(4) The image resolution is too low, it should be improved.

Reviewer #2: The authors analyzed how the choice of modelling approach, in particular how voluntary behavior change is accounted for, affect the intervention effect estimation. They conducted the analysis by experimenting different modelling methods on a same data set composed of the 500 most infected U.S. counties. They also compared the most frequently used methods from the two classes of modelling approaches, and find that computational methods that do not account for voluntary behavior changes are likely to produce larger estimates of intervention effects as assumed. In contrast, natural experimental methods are more likely to extract the true effect of interventions by ruling out simultaneous behavior change. Among different difference-in-difference estimators, the two-way fixed effect estimator seems to be an efficient one.

Totally speaking, their methods are performed appropriately and rigorously and the results seem reliable. I support its acceptance before the following issues are addressed.

1. Page 2, line 60, the authors wrote that “The modeling approaches…that do not ”. Here, what the authors said modelling approaches are all related to statistical methods. In fact, the modelling approaches are not restricted to statistics, the mathematical model, for example, SIR, SEIR or other models, act as a useful tool and has been widely studied. One can refer to these closely related books. ” Tanimoto, Jun. "Evolutionary games with sociophysics." Evolutionary Economics (2019).” And “Sun, Gui-Quan, Marko Jusup, Zhen Jin, Yi Wang, and Zhen Wang. "Pattern transitions in spatial epidemics: Mechanisms and emergent properties." Physics of life reviews 19 (2016): 43-73.”

2. The presented figures are not clear, please revise the figures to make it clearer. For example, the resolution of fig.1 is too low, and readers cannot capture the main results for the current version. Please revise it. Besides, the figure caption in fig.1, the authors simply explain fig.1A as ”Dynamics of daily R_t in the sample countries”. What is R_t? Please note that not all the readers have enough patient to read the whole text. Please revise the figure captions in a more detailed manner and conclude the main conclusions of each figure.

6. PLOS authors have the option to publish the peer review history of their article (what does this mean?). If published, this will include your full peer review and any attached files.

Reviewer #1: No

Reviewer #2: No

---

## [Author Response · Author response to Decision Letter 0]

15 Aug 2022

We thank the reviewers for their helpful suggestions. All the concerns have been addressed as follows.

Reviewer #1: By testing three major modeling methods, they demonstrate that quite different estimates of intervention effects could be acquired with different methods. The idea of the model seems somehow convincing and worth reporting. However, the way the current version of the manuscript is presented cannot be accepted and needs a major revision. Here are a couple of points that I think should be considered.

Critical remarks:

Comment 1: The model depiction is not clear to me. The language in explaining the model should be improved. In particular, I cannot find how to get indicators ‘shared change of mobility’ and ‘value of the mobility indicator in county c on day t.

Reply: The two phrases are revised to ‘γt denotes day t, representing the common part of mobility change across counties’ and ‘Google mobility indicator in county c on day t’ (line 178, 176). We also make a few other revisions to the language to make the description clearer.

(2) In Experiment 1, I suggest the authors give some details in the appendix. At least, the algorithm and the details should be given. In Experiment 2 and Experiment 3, the process of solving is not clear.

Reply: For Experiment 1, the main algorithms are provided in the Method section, which are taken from the original study of Brauner et al. (2021). Since full details of the method are provided in their original paper, we refer readers to that paper if they need further information on the method (line 208-209). In Experiment 2 and 3, OLS is used in solving the parameters. We add this information and refer readers to relevant econometric books and papers in the revised manuscript (line 222-223, 261).

(3) The reason why the authors choose these methods is not clear. In reference to what you give, their topics are all related to COVID-19. Thus, the innovative point of this paper should be emphasized. What is the difference between your content and theirs that should be given?

Reply: We chose these methods because they are the most commonly used in estimating COVID-19 intervention effects. Table S1 summarizes the existing studies on COVID-19 intervention effects when we performed the analysis. It shows that difference-in-difference (including two-way fixed effect model) and Bayesian model are the two most used methods. Therefore, we take Bayesian model and two difference-in-difference methods as the methods to compare. The innovation of our paper lies in that we focus on methodological comparison, in other words, how the choice of modelling approach would affect the estimation of COVID-19 intervention effects. On the other hand, most existing studies focus on the intervention effect itself. This point is more clearly demonstrated in the revised manuscript (line 51-52).

(4) The image resolution is too low, it should be improved.

Reply: We will follow the journal’s requirements to make sure that image resolutions are suitable for publication.

Reviewer #2: The authors analyzed how the choice of modelling approach, in particular how voluntary behavior change is accounted for, affect the intervention effect estimation. They conducted the analysis by experimenting different modelling methods on a same data set composed of the 500 most infected U.S. counties. They also compared the most frequently used methods from the two classes of modelling approaches, and find that computational methods that do not account for voluntary behavior changes are likely to produce larger estimates of intervention effects as assumed. In contrast, natural experimental methods are more likely to extract the true effect of interventions by ruling out simultaneous behavior change. Among different difference-in-difference estimators, the two-way fixed effect estimator seems to be an efficient one.

Totally speaking, their methods are performed appropriately and rigorously and the results seem reliable. I support its acceptance before the following issues are addressed.

Reply: We thank the reviewer for the positive comments. All of the issues of concern are addressed as suggested by the reviewer.

Comment 1: Page 2, line 60, the authors wrote that “The modeling approaches…that do not ”. Here, what the authors said modelling approaches are all related to statistical methods. In fact, the modelling approaches are not restricted to statistics, the mathematical model, for example, SIR, SEIR or other models, act as a useful tool and has been widely studied. One can refer to these closely related books. ” Tanimoto, Jun. "Evolutionary games with sociophysics." Evolutionary Economics (2019).” And “Sun, Gui-Quan, Marko Jusup, Zhen Jin, Yi Wang, and Zhen Wang. "Pattern transitions in spatial epidemics: Mechanisms and emergent properties." Physics of life reviews 19 (2016): 43-73.”

Reply: Many thanks for suggesting the literature. The statement is revised to “While mathematical models (such as SIR and SEIR models) are classical methods in modelling epidemics (9, 10), statistical models and their combinations are widely applied in estimating the effects of interventions. Relevant statistical models employed by existing analyses can be grouped into two classes…” (line 63-65).

Comment 2. The presented figures are not clear, please revise the figures to make it clearer. For example, the resolution of fig.1 is too low, and readers cannot capture the main results for the current version. Please revise it. Besides, the figure caption in fig.1, the authors simply explain fig.1A as ”Dynamics of daily R_t in the sample countries”. What is R_t? Please note that not all the readers have enough patient to read the whole text. Please revise the figure captions in a more detailed manner and conclude the main conclusions of each figure.

Reply: Fig. 1 is compressed in the process of building the submission file, though we tried to use a higher resolution figure. We will follow the journal’s requirements to make sure that image resolutions are suitable for publication. And all figure captions are revised as suggested by the reviewer.

---

## [Decision Letter · Decision Letter 1]

17 Oct 2022

Comparing modelling approaches for the estimation of government intervention effects in COVID-19: Impact of voluntary behavior changes

PONE-D-22-07518R1

Dear Dr. Wang,

We’re pleased to inform you that your manuscript has been judged scientifically suitable for publication and will be formally accepted for publication once it meets all outstanding technical requirements.

Kind regards,

Lei Shi

Academic Editor

PLOS ONE

Additional Editor Comments (optional):

Reviewers' comments:

Reviewer's Responses to Questions

**Comments to the Author**

1. If the authors have adequately addressed your comments raised in a previous round of review and you feel that this manuscript is now acceptable for publication, you may indicate that here to bypass the “Comments to the Author” section, enter your conflict of interest statement in the “Confidential to Editor” section, and submit your "Accept" recommendation.

Reviewer #1: All comments have been addressed

Reviewer #2: All comments have been addressed

2. Is the manuscript technically sound, and do the data support the conclusions?

Reviewer #1: Yes

Reviewer #2: Yes

3. Has the statistical analysis been performed appropriately and rigorously? 

Reviewer #1: Yes

Reviewer #2: Yes

4. Have the authors made all data underlying the findings in their manuscript fully available?

Reviewer #1: No

Reviewer #2: Yes

5. Is the manuscript presented in an intelligible fashion and written in standard English?

Reviewer #1: Yes

Reviewer #2: Yes

6. Review Comments to the Author

Reviewer #1: Thanks to the author for answering my question. It would be better if the data links were presented.

Reviewer #2: (No Response)

7. PLOS authors have the option to publish the peer review history of their article (what does this mean?). If published, this will include your full peer review and any attached files.

Reviewer #1: No

Reviewer #2: No

---

## [Editor Report · Acceptance letter]

8 Nov 2022

PONE-D-22-07518R1 

Comparing modelling approaches for the estimation of government intervention effects in COVID-19: Impact of voluntary behavior changes 

Dear Dr. Wang:

I'm pleased to inform you that your manuscript has been deemed suitable for publication in PLOS ONE. Congratulations! Your manuscript is now with our production department. 

Kind regards, 

on behalf of

Dr. Lei Shi 

Academic Editor

PLOS ONE